# Defense Response to *Hemileia vastatrix* in Susceptible Grafts onto Resistant Rootstock of *Coffea arabica* L.

Edgar Couttolenc-Brenis [1,2], Gloria Carrión [3], Luc Villain [4,5], Fernando Ortega-Escalona [6], Martín Mata-Rosas [1,*] and Alfonso Méndez-Bravo [7,*]

1. Red de Manejo Biotecnológico de Recursos, Instituto de Ecología, Xalapa 91073, Mexico; edg.couttolenc@gmail.com
2. Instituto Nacional de Investigaciones Forestales Agrícolas y Pecuarias, Cotaxtla 94283, Mexico
3. Red de Biodiversidad y Sistemática de Hongos, Instituto de Ecología, Xalapa 91073, Mexico; gloria.carrion@inecol.mx
4. CIRAD, UMR DIADE, F-34394 Montpellier, France; luc.villain@cirad.fr
5. DIADE, Université de Montpellier, CIRAD, IRD, F-34090 Montpellier, France
6. Red de Ecología Funcional, Instituto de Ecología, Xalapa 91073, Mexico; fernandoortegaescalona57@gmail.com
7. CONACYT-Escuela Nacional de Estudios Superiores Unidad Morelia, Laboratorio Nacional de Análisis y Síntesis Ecológica, Universidad Nacional Autónoma de México, Morelia 58190, Mexico
* Correspondence: martin.mata@inecol.mx (M.M.-R.); amendezbravo@enesmorelia.unam.mx (A.M.-B.)

**Abstract:** The use of resistant cultivars and fungicides are common methods to control coffee leaf rust (CLR), the main disease that affects the Arabica coffee crop. In this study, we evaluated the response of grafted and ungrafted plants during the early stage of *Hemileia vastatrix* infection. We used ungrafted plants of Oro Azteca (resistant cultivar) and Garnica (susceptible cultivar), and grafted plants, combining both as rootstock and graft (Garnica/Oro Azteca and Oro Azteca/Garnica). All plants were inoculated with *H. vastatrix* uredospores, and we quantified the development of fungal structures in the leaf tissue of inoculated plants using qRT-PCR to measure relative expression of two pathogenesis recognition genes (*CaNDR1* and *CaNBS-LRR*) and three genes associated with the salicylic acid (SA) pathway (*CaNPR1*, *CaPR1* and *CaPR5*). In Garnica grafted on Oro Azteca, the fungal structures recorded were significantly less than in Garnica ungrafted plants. In addition, the expression of defense-related genes in grafted plants was higher than in ungrafted plants. Our results indicate that the defense response to CLR is strongly influenced by the rootstock employed.

**Keywords:** grafting effect; defense-related gene expression; leaf rust; coffee

## 1. Introduction

Coffee is one of the most important agricultural sources of foreign income in Tropical and Subtropical countries [1]. Therefore, controlling and managing diseases and pests is key to preserving the income of producers that depend on this crop. Coffee leaf rust (CLR) is one of the diseases that has had the most impact on global coffee production in recent years by reducing its yield, particularly in Latin America [2–5].

CLR is caused by the obligate pathogen *H. vastatrix* [6,7], which penetrates into the leaf mesophyll through the stoma [8]. Once established in the leaf tissues, the fungus generates chlorotic spots that become orange when completing its developmental stage, due to the production of spores [4,5]. At low infestation levels, the effects of CLR on the development and productivity of the plant remain small; however, when infestation levels rise, it causes severe defoliation of the plants, thus reducing its yield and in some cases, the death of the plant [9].

The most common control method for CLR has been based on chemicals, by applying systemic fungicides (triazole and strobilurin) and contact treatments mainly based on copper applications [5,9–11]. The management of shade, fertilization and the fruiting

load, among other agronomic practices, can reduce the severity of the infection [12–14]. Some mycoparasites have also been identified (by testing them in vitro) as a potential source for the biological control of rust. Among the fungi that stand out are the genus *Lecanicillium*, *Calcarisporium*, *Sporthrix* and *Simplicillium* [15–17]. Bacteria belonging to the genus *Bacillus* and *Pseudomonas* genera have been evaluated [16,18,19]. However, the use of these biological control agents for CLR is not widespread [2,5].

In addition to the mentioned applications of chemical and biological agents and the agronomic practices, the use of resistant cultivars has been more effective in CLR management. Most of the resistant cultivars have been obtained from a natural interspecific hybrid of *C. arabica* L. and *C. canephora* L., which was identified in Timor Island (HdT) [4]. However, many cultivars derived from the HdT present a lower organoleptic quality, compared to the traditional susceptible varieties [20–22]. Additionally, in recent years, the breakdown of resistance in some cultivar derived from HdT has been observed [23].

An alternative that would allow the preservation of susceptible materials that confer high cup quality is grafting these susceptible cultivars onto CLR-resistant cultivars. Studies in some other crops have shown a regulating effect of the response to biotic and abiotic stress, which is conferred by the rootstock [24–28]. In fruit trees, there are some examples, such as apples (*Malus pumila* L.), where changes in the severity of infections caused by *Erwinia amylovora* in the cv. Gala graft were observed and associated with the rootstock cultivar [29,30]. In grafts of the Sonora almond cultivar (*Prunus dulcis* L.), the severity of the infection caused by *Xylella fastidiosa* is related to the rootstock used [31]. Similar results have also been reported for vegetables. For example, when grafted onto *Capsicum annuum* var. *cerasiforme*, *C. annuum* shows more resistance to powdery mildew [32]. *Cucumis* is another genus that showed that the type of rootstock influenced the response to a leaf pathogen. Gu et al. [33] observed a reduction in the severity of the infection by *Pseudoperonospora cubensis*, which causes downy mildew in cucumbers (*Cucumis sativus* L.), when grafted onto chilacayote (*Cucurbita ficifolia* L.). Moreover, studies with citrus, Meliaceae species and *Fraxinus,* show that grafting differentially impacts the plant response to the attack by insects, depending on the rootstock employed [34–37].

The mechanisms that determine the interaction between the rootstock and the graft have not yet been completely elucidated [38]. Some studies show the translocation of secondary metabolites from the rootstock to the graft [39]. Research has even identified that RNA and mRNA fragments are transported through the vascular system [40–42]. It has also been observed that the rootstock modifies the expression of genes associated with the stress response in the scion of the vine (*Vitis* sp.), apple (*M. pumila* L.), tangerine (*Citrus unshiu*) and eggplant (*Solanum melongena* L.), among others [29,43–46]. Therefore, grafting would be a key point for integral crop protection.

In coffee, grafting on *C. canephora* rootstocks has conventionally been used to manage diseases caused by nematodes [47–50]. Moreover, no significant effect of grafting has been observed on the behavior of grafted coffee trees and on the *C. arabica* scion coffee cup quality [21,51]. However, no information exists about the effect of using resistant rootstocks on foliar pathogens, such as *H. vastatrix*.

Thus, our objective was to evaluate the effect of the *C. arabica* CLR-resistant cultivar Oro Azteca [52] as rootstock on the response of the CLR infection in the *C. arabica* scion cv. Garnica, a susceptible cultivar, and to determine the potential of grafting as a tool for the control of *H. vastatrix*. In order to identify whether there was a differential response to the attack by *H. vastatrix*, the development of fungal structures on the foliar tissue of the grafted plants and the expression of defense genes involved in the response to this pathogen were evaluated.

## 2. Materials and Methods

### 2.1. Experimental Design and Plant Material

This study was carried out under four conditions of grafting: (1) Garnica plants without grafting; (2) Oro Azteca plants without grafting; (3) Grafted Garnica/Oro Azteca

plants; and (4) Grafted Oro Azteca/Garnica plants. Each condition included 24 repetitions (plants), distributed through an experimental design in random blocks.

For the susceptible material, we used the Garnica variety which was developed by the Instituto Mexicano del Café from the cross between Caturra Amarillo and Mundo Novo [53]; for the resistant material we used the Oro Azteca variety, developed by the Instituto Nacional de Investigaciones Forestales Agrícolas y Pecuarias (INIFAP) from the cross between the Timor hybrid 832/1 and Caturra [52,54]. Plants were produced from seeds in polyethylene bags in the nursery of the Centro Internacional de Capacitación en Cafeticultura y Desarrollo Sustentable A.C. (CICADES) until they had two pairs of leaves. The scion were grafted onto rootstock seedlings at the cotyledon stage, using the Reyna method (hypocotyledonary grafting; Figure S1) [51]. Six months after grafting, plants were used for the pathogenesis assays.

### 2.2. Preparation of the Uredospore Inoculum

The uredospores of *H. vastatrix* were obtained by collecting CLR-infested leaves from plantations of *C. arabica* cv. Typica in Huatusco, Ixhuatlán del Café and Zentla municipalities, Veracruz State. Uredospores were collected from ten plants in each of the selected orchards, then isolated under a stereoscope microscope (Leica SE6, Wetzlar, Germany) and placed in 1.5 mL microtubes. The inoculum was prepared in a sterile distilled water solution with a $1.5 \times 10^{-5}$ spores mL$^{-1}$ concentration and a drop of Tween® 80 at 0.01%.

### 2.3. Inoculation and Incubation

The *H. vastatrix* spore suspension was sprayed on the first and second pair of mature leaves on the abaxial surface [55], distributing the inoculum throughout all of the surface. Plants were placed in an incubation chamber under (80% HR) humidity conditions, with an average temperature of 22 °C. Plants were initially kept in the dark for 72 h to help the germination of the uredospores. Afterward, plants were placed in a greenhouse with a 12 h photoperiod.

### 2.4. Observation of the Development of Hemileia vastatrix during the First Stages of Infection in the Foliar Tissue

Post-infection evaluation of *H. vastatrix* consisted of quantifying the proliferation of fungal structures on vegetal tissue, through observations in a bright-field optical microscope (Leica DM750 M, Wetzlar, Germany) with a 50X objective, comparing the presence of penetration hyphae and anchors, haustorial mother cells (HMC) and haustoria in the four grafting conditions. Therefore, samples of three control or inoculated plants were collected at 2, 6, 12, 24, 48 and 72 h after inoculation (hai) to obtain histological preparations. Samples were placed in plant fixative FAA (Formaldehyde:Alcohol:Acetic Acid, 10:50:5 + 35% water) to include them in paraffin. Transversal cuts 18 μm thick were made with a microtome (Leica Biosystems RM2125 RTS, Nussloch, Germany) [56,57]. The sections obtained were dyed through immersion in a solution of lactophenol cotton blue 0.05% for two minutes. Afterward, they were rinsed with a lactophenol solution and mounted in a polyvinyl alcohol solution. Observations and recording of data were made for each of the observation times. The data on percentages of the presence for each fungus development stage of the infection were calculated in relation to the total of the observed stomata, and transformed through the arccosine function. A variance analysis was performed using the university version of the SAS© GLM procedure (SAS University Edition, SAS Institute, NC, USA) software, and a comparison of means was conducted using the SAS©LSMEANS (SAS University Edition, SAS Institute, NC, USA) procedure. The data analysis was conducted for the presence of fungal structures per observation time, and for the total of detected structures in every graft condition.

### 2.5. Selection of Target Genes

Five genes reported as associated with the response to *H. vastatrix* infection in *C. arabica* were selected [23,58–60]. Furthermore, two constitutive genes for coffee [61–63] were selected. The oligonucleotides specific to these genes were designed using the Primer3Plus® [64] software, selecting a unique sequence for each gene obtained from the NCBI database. The parameters used for designing these oligonucleotides were 20 base pairs per oligo at a 60 °C maximum melting temperature, and a > 40% G-C relation. Oligos were synthesized at the T4Oligo laboratory in Irapuato, Mexico (Table 1).

**Table 1.** Primers designed for genes *NBS-LRR, NDR1b, NPR1, PR1, PR5, UbiE2* and *CaGAPH*, used in the expression analysis for qRT-PCR.

| Gene | Primer | Gene Type |
|---|---|---|
| *CaNBS-LRR* | F5′-CCAAAAACTTTgggTTggTg-3′<br>R5′-TCCATTgCATTCTCATCTg-3′ | Plant defense involved in pathogen recognition |
| *CaNDR1b* | F5′-CTTACAgggCggTgTCAAAT-3′<br>R5′-TACCACTAgCCCAggACAgC-3′ | Plant defense, hypersensitive response |
| *CaNPR1* | F5′-gACgCTgCAgTgAAgAAAC-3′<br>R5′-TgATAgCTTCCCAggCATCT-3′ | Plant defense involved in the salicylic acid pathway |
| *CaPR1* | F 5′-CaggAATgCgggCATTATAC-3′<br>R 5′-CAATCgCATgggTTTgATAA-3′ | Plant defense involved in the synthesis of salicylic acid |
| *CaPR5* | F 5′-CtgCCTgAgTTgCAgCAATA-3′<br>R 5′-TTTCCCTTgTTgATggCTTC-3′ | Plant defense involved in the salicylic acid pathway |
| *CaUbiE2* | F5′-CCATTTAAACCCCCAAAggT-3′<br>R5′-ggTCCAgCTTCgAgCAgTAg-3′ | *C. arabica* constitutive |
| *CaGAPH* | F5′- gCAgCACTTCATggTTCTgA-3′<br>R5′-TTTCCACATTTCAgCCCTTC-3′ | *C. arabica* constitutive |

### 2.6. Gene Expression Analysis

#### 2.6.1. RNA Extraction

Three plants for each sampling time were selected (0, 2, 6, 12, 24, 48, 72 hai) from the 24 plants by treatment (Garnica, Garnica/Oro Azteca, Oro Azteca, Oro Azteca/Garnica). Three leaves were collected from each plant, immersed in liquid nitrogen and grounded in porcelain mortars. The RNEasy kit from Qiagen (Hilden, Germany) was used to perform the extraction, following the procedure indicated by the manufacturer.

#### 2.6.2. qRT-PCR: Synthesis of the First cDNA Chain

The first cDNA chain was synthesized starting at 30 μg total RNA. Each reaction mix contained: 1.0 μg/mL total RNA, a solution of the first chain 10 X, 25 mM MgCl2, 10 mM dNTPs, 40 unit/mL RNasin Inh, 0.5 mg/mL oligo(dT) and 25 units/mL reverse transcriptase SuperScript III (Invitrogen®, Waltham, Massachusetts, USA). For the synthesis of the second chain, the amplification conditions were: 70 °C for 10 min and 42 °C for 2 h.

#### 2.6.3. qRT-PCR Analysis

Each reaction was completed using 3 μL cDNA, 1 X reaction mix (20 μL) "SYBR Green PCR Master Mix (Applied Biosystems) and 500 nM of each of the primers corresponding to each gene. The amplification conditions were: 94 °C for 10 min, 40 cycles at 94 °C for 30 s, 60 °C for 30 s and 72 °C for 40 s. The amplifications for qRT-PCR were conducted with a thermocycler "7500 Fast Real-Time PCR System (Applied Biosystems, Foster City, CA, USA)". The relative quantification of the abundance of each transcript was calculated and normalized, concerning the average of the constitutive genes *CaUbiE2* and *CaGAPH* [59,62,65] of the Garnica and Oro Azteca ungrafted plants. Four independent replicas were obtained at less than 0.1 standard error for each of the samples; each

expression value is the average of these replicas. Calculations were performed using "7500 Software v2.0.1 (Applied biosystems, Foster City, CA, USA)" and through applying the $2^{-\Delta\Delta CT}$ method [66,67]. Amplification efficiency for each set of oligonucleotides was determined by performing a series of dilutions (1:5), while the specificity of amplifications was calculated by dissociation curves, obtaining fluorescence values between 65 °C and 95 °C. On average, amplification curves were quantified starting on cycle 15 of each sample.

2.6.4. Quantification of Relative Gene Expression

The calculations of relative expression values were obtained by comparing the threshold value of the cycle (Ct values) of each gene studied and the reference genes when there exists a PCR reaction efficiency of 100%. The average Ct values for *CaUbiE2* and *CaGAPH* genes were used to calibrate the expression level of the target genes. Hence, ΔΔCT was obtained from the ΔCT (of each sample tested)—ΔCT (0 hai), which represents ΔCT (CT (target)* E)−(CT (control) * E), and E is the efficiency of estimated PCR E = 10 ˆ (−1/k)) −1). From this calculation, the relative expression value $2^{-\Delta\Delta CT}$ was obtained, and it was used to quantify the transcript level at different times after inoculation with *H. vastatrix*, following the formula proposed by Livak et al., [66]:

$$\Delta\Delta C_T = (C_{T.Target} - C_{T.\ Control})_{Time\ X} - (C_{T.\ Target} - C_{T.\ Control})_{Time\ 0}$$

$$Target = Expression\ of\ coffee\ defense\ genes\ (CaNBS - LRR,\ CaNDR1b,\ CaNPR1,\ CaPR1,\ CaPR5)$$

$$Control = Average\ of\ expression\ of\ coffee\ housekeeping\ genes\ CaUbi2\ and\ CaGAPDH$$

Once the expression values were normalized based on the constitutive genes, we graphed each gene's value for different conditions. The resulting expression levels for each gene were compared using a Kruskal–Wallis non-parametric test in Infostat® (Nai-robi, Kenya).

**3. Results**

*3.1. Development of Hemileia vastatrix in Infected Leaves*

The infection of coffee leaves by *H. vastatrix* begins with the uredospore germination and the subsequent appressoria formation over the stomata, followed by the development of penetration hyphae into the stomatic chambers. From each penetration hypha, two anchor-like branches are formed, with each branch giving rise to a structure known as the haustorial mother cell (HMC), from which the haustoria develop [4,5]. Disease progression and the development of fungal structures varied, depending on the combination of grafting and the evaluation time. Ungrafted Garnica plants showed a higher proliferation of penetration hyphae from 2 to 6 hai, compared to the other grafting combinations, reaching 42% of infection points with penetrating hyphae in this phase (Table 2). From this point, Garnica showed the highest presence of anchor structures at 12 hai, reaching its maximum at 72 hai. In almost all sample times, Garnica had the greater value of anchor percentage per infection point. The grafts and the ungrafted Oro Azteca showed a similar percentage of anchors (Figure 1a, Table 2).

The presence of haustorial mother cells (HMC) ranged from 0 to 0.3% during the first 48 hai in all conditions (Figure 1b), and increased at 72 hai, especially in the leaves of un-grafted Garnica, which showed the highest presence of HMC reaching 6%. Simultaneously, in the other conditions, the formation of this structure was similar at percentages of less than 2% (Table 2; Figure 1b).

Quantification of fungal structures in the leaves of the grafts Garnica/Oro Azteca and Oro Azteca/Garnica only showed significant differences among them at 12 hai ($p < 0.0001$). At this time, Garnica/Oro Azteca plants had more haustoria than Oro Azteca/Garnica, reaching its maximum recorded haustoria, with 23% of penetration points. In the remaining times analyzed, the trend between grafts were similar, with less than 1% difference in the percentage of accounted haustoria per infection points (Figure 1c; Table 2).

**Table 2.** Statistical values of fungal structure proliferation in inoculated leaves of ungrafted and grafted plants.

| | Hai | Pr > F | Garnica | | Garnica/Oro Azteca | | Oro Azteca | | Oro Azteca/Garnica | |
|---|---|---|---|---|---|---|---|---|---|---|
| | | | μ | S.E. | μ | S.E. | μ | S.E. | μ | S.E. |
| **PH** | 2 | 0.0910 | 0.81 a | 0.48 | 0 a | 0.00 | 0 a | 0.00 | 0.26 a | 0.26 |
| | 6 | <0.0001 | 41.64 a | 2.42 | 18.25 b | 1.55 | 5.78 c | 0.88 | 3.57 c | 0.78 |
| **Anchor** | 6 | <0.0001 | 2.68 a | 0.69 | 0.75 b | 0.34 | 0.17 b | 0.16 | 0 b | 0 |
| | 12 | <0.0001 | 7.66 a | 1.27 | 1.62 b | 0.4 | 1.24 b | 0.41 | 1.17 b | 0.53 |
| | 24 | 0.0018 | 5.06 a | 1.14 | 1.67 b | 0.6 | 2.37 b | 0.75 | 2.75 ab | 0.78 |
| | 48 | 0.3212 | 2.29 a | 0.72 | 3.41 a | 0.92 | 1.53 a | 0.7 | 3.58 a | 0.9 |
| | 72 | <0.0001 | 9.49 a | 1.47 | 2.58 b | 0.8 | 3.37 b | 1.05 | 2.81 b | 0.78 |
| **HMC** | 12 | 0.0940 | 0.20 a | 0.22 | 0 a | 0 | 0 a | 0 | 0 a | 0 |
| | 24 | 0.3329 | 0 a | 0 | 0 a | 0 | 0.25 a | 0.26 | 0 a | 0 |
| | 48 | – | 0 | 0 | 0 | 0 | 0 | 0 | 0 | 0 |
| | 72 | 0.0003 | 6.33 a | 1.16 | 2.32 b | 0.76 | 2.07 b | 0.81 | 2.04 b | 0.65 |
| **Haustoria** | 12 | <0.0001 | 34.88 a | 2.23 | 22.87 b | 2.02 | 11.88 c | 1.40 | 14.22 c | 1.70 |
| | 24 | <0.0001 | 21.52 a | 2.11 | 9.38 b | 1.41 | 13.51 b | 2.02 | 10.50 b | 1.48 |
| | 48 | 0.0780 | 13.23 a | 1.61 | 16.34 a | 1.85 | 10.94 a | 1.65 | 17.39 a | 2.02 |
| | 72 | <0.0001 | 23.36 a | 2.11 | 4.64 b | 1.04 | 5.44 b | 1.24 | 4.08 b | 0.95 |

Different letters within the same line indicate significant differences between treatments in multiple comparisons of Tukey–Kramer ($\alpha$ = 0.05); (PH) penetration hyphae, (HMC) haustorial mother cells, (hai) hours after inoculation.

Of all the conditions studied, the ungrafted Oro Azteca showed the least presence of haustoria at almost all sampling times, showing the greatest percentage of 14% at 24 hai. It is important to note that haustoria detection was similar in the graft Garnica/Oro Azteca and in the ungrafted Oro Azteca plants after 24 hai (Figure 1c; Table 2).

When making a global comparison of the four evaluated conditions, highly significant differences were found in the percentage of anchors ($p$ < 0.0001), HCM ($p$ < 0.0001) and haustoria ($p$ < 0.0001). The highest anchor detection occurred in Garnica (5%), and it was statistically different from the other treatments. The presence of HMC showed a similar abundance to anchors, and Garnica had the highest percentage compared to the other three treatments at 1% (Figure 2). The presence of haustoria recorded the highest percentage in Garnica (15%), a second group was formed by the grafts Garnica/Oro Azteca with a mean of 8%, and, lastly, Oro Azteca and Oro Azteca/Garnica formed a single group with 7% presence of haustoria (Figure 2). The number of fungal structures registered during the time course of the infection for each plant condition was statistically analyzed, showing that the anchors' abundance only differed significantly in ungrafted Garnica, whilst differences in HCM and haustoria were highly significant in all conditions (Tables S1 and S2).

The long-term effects of disease progression on ungrafted and grafted plants were monitored by visually recording evident symptoms and signs of CLR at 30, 60, 100, 130, 165, 208 and 236 days after infection (dai) (Table S3; Figure S2). Leaves from ungrafted Garnica and Garnica/Oro Azteca grafted plants showed chlorotic spots and pustules at 30 dai, while the leaves from ungrafted Oro Azteca displayed moderate signs until 165 dai (Table S3).

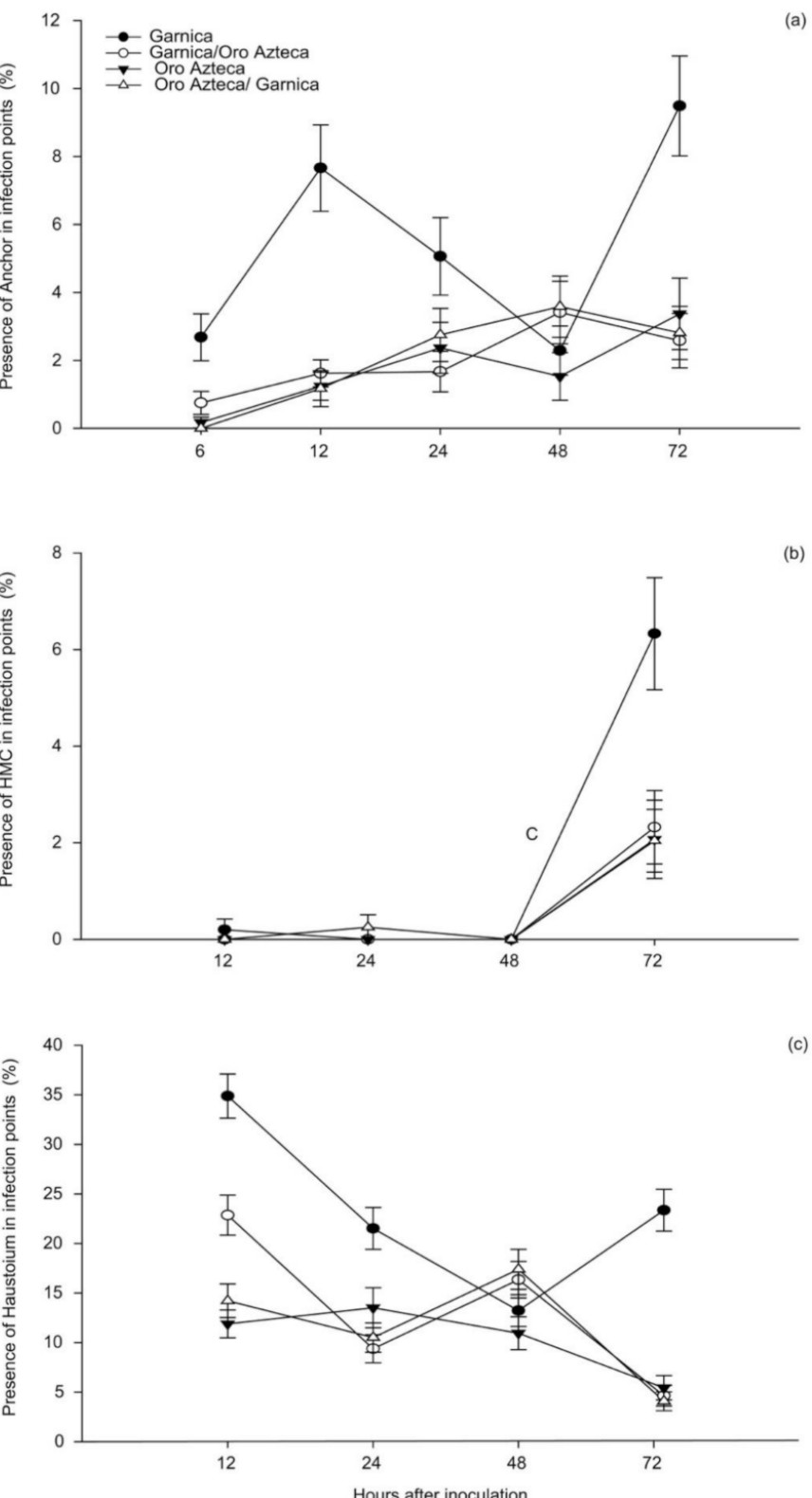

**Figure 1.** Presence of fungal structures. (**a**) Anchors, (**b**) haustorial mother cells (HMC) and (**c**) haustoria. The abundance was estimated as the percentage of infection points (stomatic chambers), in which the presence of fungal structures were observed in different periods sampled during 72 h after inoculation (hai) in grafted and ungrafted plants.

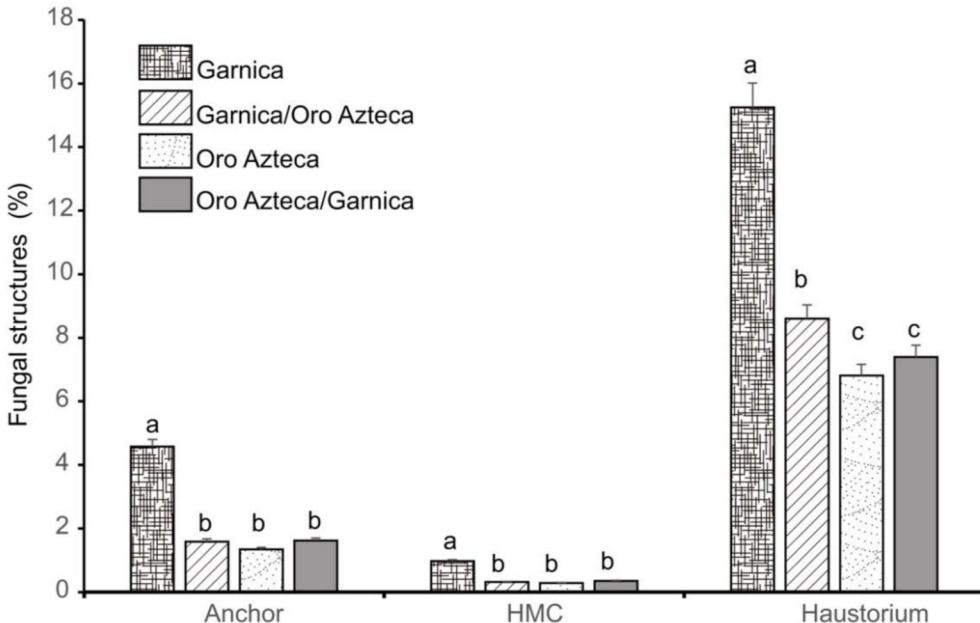

**Figure 2.** Global comparison of the presence of fungal structures (anchors, haustorial mother cells and haustoria) in each of the conditions evaluated; (HCM) haustorial mother cells. Different letters indicate significant differences by a Tukey-Kramer adjustment ($\alpha$ = 0.05).

### 3.2. Expression of CLR Recognition Genes

The Kruskal–Wallis non-parametric tests showed that in all the sampling times there were significant differences in the differential expression of the two pathogen recognition genes (Table 3). Between the ungrafted plants (Garnica and Oro Azteca) and grafted plants (Garnica/Oro Azteca and Oro Azteca/Garnica), the expression of gene *CaNDR1b* were significantly different ($\alpha$ = 0.5, Tukey–Kramer Adjustment) in most of the sampling times, except at 96 hai. The ungrafted Oro Azteca had a lower gene *CaNDR1b* expression level than the control at 2 hai. Garnica showed lower gene *CaNDR1b* expression levels than the control at 6, 72 and 96 hai, and was similar at 24 hai. In the case of grafts in both conditions, the expression levels of *CaNDR1b* gene were consistently higher than the control. It stood out that the level of expression of the *CaNDR1b* gene in Garnica/Oro Azteca was higher compared to the other treatments at 6 (1328 fold change), 12 (942 fold change) and 48 hai (249 fold change). The level of expression of *CaNDR1b* gene in the Oro Azteca/Garnica graft was higher at 2 (629 fold change) and 24 hai (251 fold change). Meanwhile, the ungrafted Oro Azteca only showed a larger fold change of *CaNDR1b* gene expression at 72 hai than the rest of the evaluated conditions. At 96 hai, Oro Azteca and the grafts showed a similar *CaNDR1b* gene expression level (Figure 3a).

**Table 3.** Values obtained from the Kruskal–Wallis test comparing the relative expression of CLR recognition genes in coffee.

| Hai | *CaNDRb1* | | | *CaNBS-LRR* | | |
|---|---|---|---|---|---|---|
| | $\chi^2$ | DF | $Pr > \chi^2$ | $\chi^2$ | DF | $Pr > \chi^2$ |
| 2 | 12.20 | 3 | 0.0067 | 14.1176 | 3 | 0.0027 |
| 6 | 14.1176 | 3 | 0.0027 | 14.1176 | 3 | 0.0027 |
| 12 | 14.1176 | 3 | 0.0027 | 13.15 | 3 | 0.0043 |
| 24 | 13.15 | 3 | 0.0043 | 13.15 | 3 | 0.0043 |
| 48 | 12.20 | 3 | 0.0067 | 14.1176 | 3 | 0.0027 |
| 72 | 14.1176 | 3 | 0.0027 | 13.15 | 3 | 0.0043 |
| 96 | 10.0792 | 3 | 0.0179 | 14.1176 | 3 | 0.0027 |

In regard to the expression levels of *CaNBS-LRR* gene, all the conditions were significantly different in the time periods evaluated. The ungrafted Oro Azteca showed the highest expression level of *CaNBS-LRR* gene in almost all the evaluated times, except at 12 hai, when the graft Garnica/Oro Azteca showed the highest expression levels of this gene. In the Garnica variety, the expression level of *CaNBS-LRR* gene was lower than in the control, at 2, 6 and 24 hai, and it was statistically similar to the control at 72 hai. In the case of grafts in both conditions, the *CaNBS-LRR* gene expression level was lower than in the control at 2, 24, 72 and 96 hai in Garnica/Oro Azteca, and in Oro Azteca/Garnica at 2, 12, 24 and 48 hai (Figure 3b).

### 3.3. Expression of the Salicylic Acid (SA) Pathway-Related Genes

The results of the Kruskal–Wallis test showed that the expression of the transcriptional regulator of the SA-signaling pathway, *CaNPR1*, was significantly different in all treatments in each of the sampled times. Garnica showed a similar level of *CaNPR1* expression, compared to the control at 24 hai (1.33 fold change) and at 48 hai (1.135 fold change). In all other time samples, this treatment had a *CaNPR1* expression level lower than the control. Oro Azteca showed *CaNPR1* higher expression levels than the control in almost all times except at 2 hai. Only the graft Garnica/Oro Azteca showed a relative expression higher than the control in all sampling times (Figure 4a). When comparing the ungrafted and grafted plants, the highest expression level of the *CaNPR1* gene occured in the grafts.

The gene *CaPR1* did not show a statistical difference between expression levels of Oro Azteca (1.56 fold change) and Oro Azteca/Garnica (1.56 fold change) at 2 hai. Similarly, at 72 hai, Oro Azteca (14.99 fold change) and Garnica/Oro Azteca (18.31 fold change) did not present significant differences in their *CaPR1* expression level. All other treatments showed differential responses in each of the periods sampled. At 2, 6, 12, 48 and 96 hai, the Garnica/Oro Azteca graft showed the highest level of *CaPR1* expression, with a 249.70 fold change at 6 hai and a 245.71 fold change at 12 hai when it reached its highest level. Oro Azteca registered its highest *CaPR1* expression level at 6 hai (25.34 fold change), and in comparison to the other treatments, it was at 24 hai that it showed its highest relative expression. The treatments that had *CaPR1* expression levels lower than the control were Garnica at 6, 12, 72 and 96 hai, and Oro Azteca/Garnica at 12, 48, 72 and 96 hai (Figure 4b).

The expression levels of the gene *CaPR5* did not show significant differences between treatments at 2 hai, while Garnica presented a 4.35 fold change and Oro Azteca a 4.72 fold change expression level. Moreover, Oro Azteca did not show differences for *CaPR5* expression level in this period with Garnica/Oro Azteca (5.98 fold change). Similarly, at 24 hai, Garnica (1.06 fold change) and Oro Azteca/Garnica (1.24 fold change) did not show significant differences for *CaPR5* expression level. Garnica showed *CaPR5* expression levels similar to the control at 24, 48 and 72 hai, and it only went lower than the control at 6 hai. Another treatment with lower expression levels of *CaPR5* than the control was Oro Azteca/Garnica at 12 and 48 hai. Both Oro Azteca and Garnica/Oro Azteca always showed higher levels of *CaPR5* expression than the control, with the highest expression levels for Oro Azteca at 24 hai (16.15 fold change) and 72 hai (249.47 fold change), when it reached its highest expression level. In Garnica/Oro Azteca, the expression level of *CaPR5* was higher than for the other treatments at 6, 12, 48 and 96 hai, with its highest level at 12 hai (596.56 fold change) (Figure 4c).

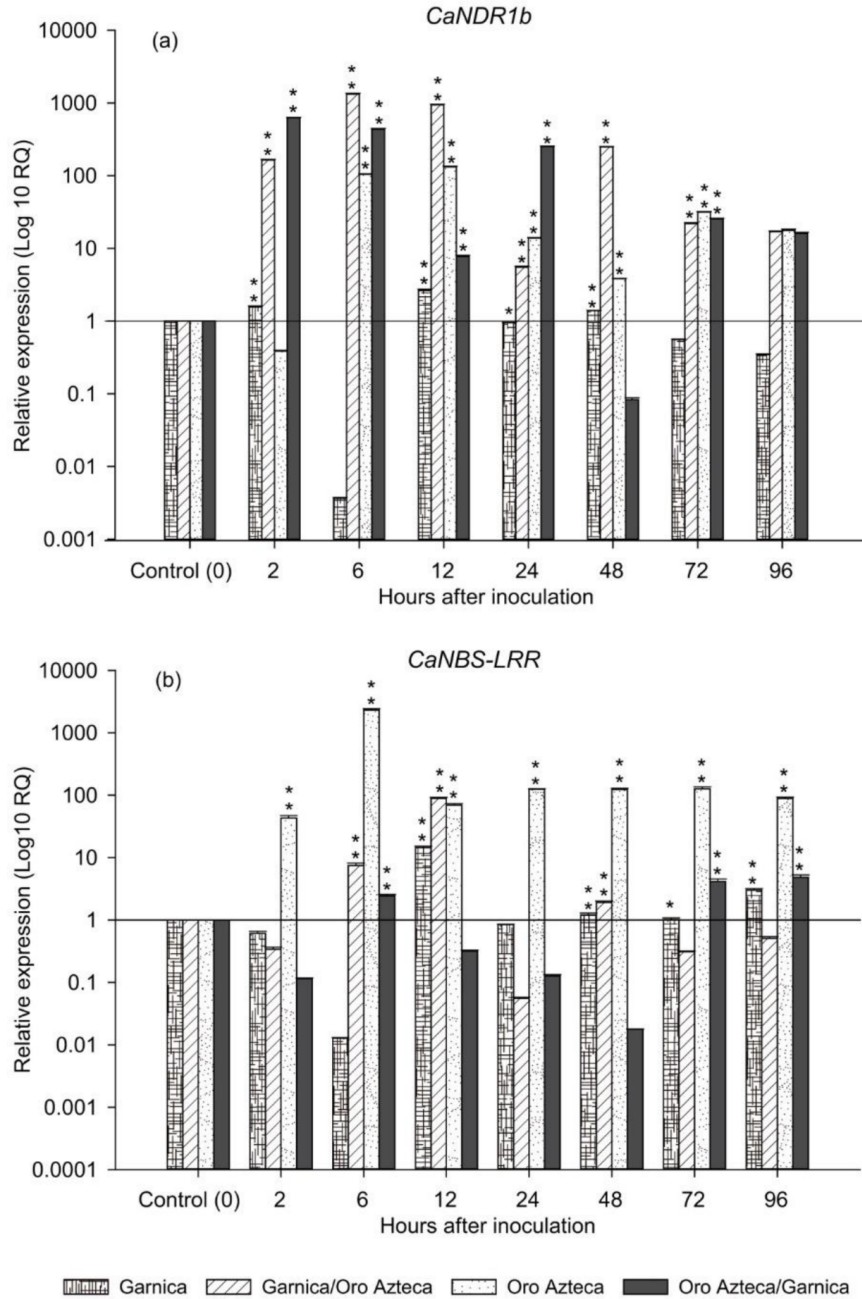

**Figure 3.** Relative expression of the pathogen recognition genes *CaNDR1b* (**a**) and *CaNBS-LRR* (**b**) in grafted and ungrafted plants during 96 hai, and using as control plants at 0 hai. * There were no significant differences between the control and the treatment. ** There were significant differences between the treatments.

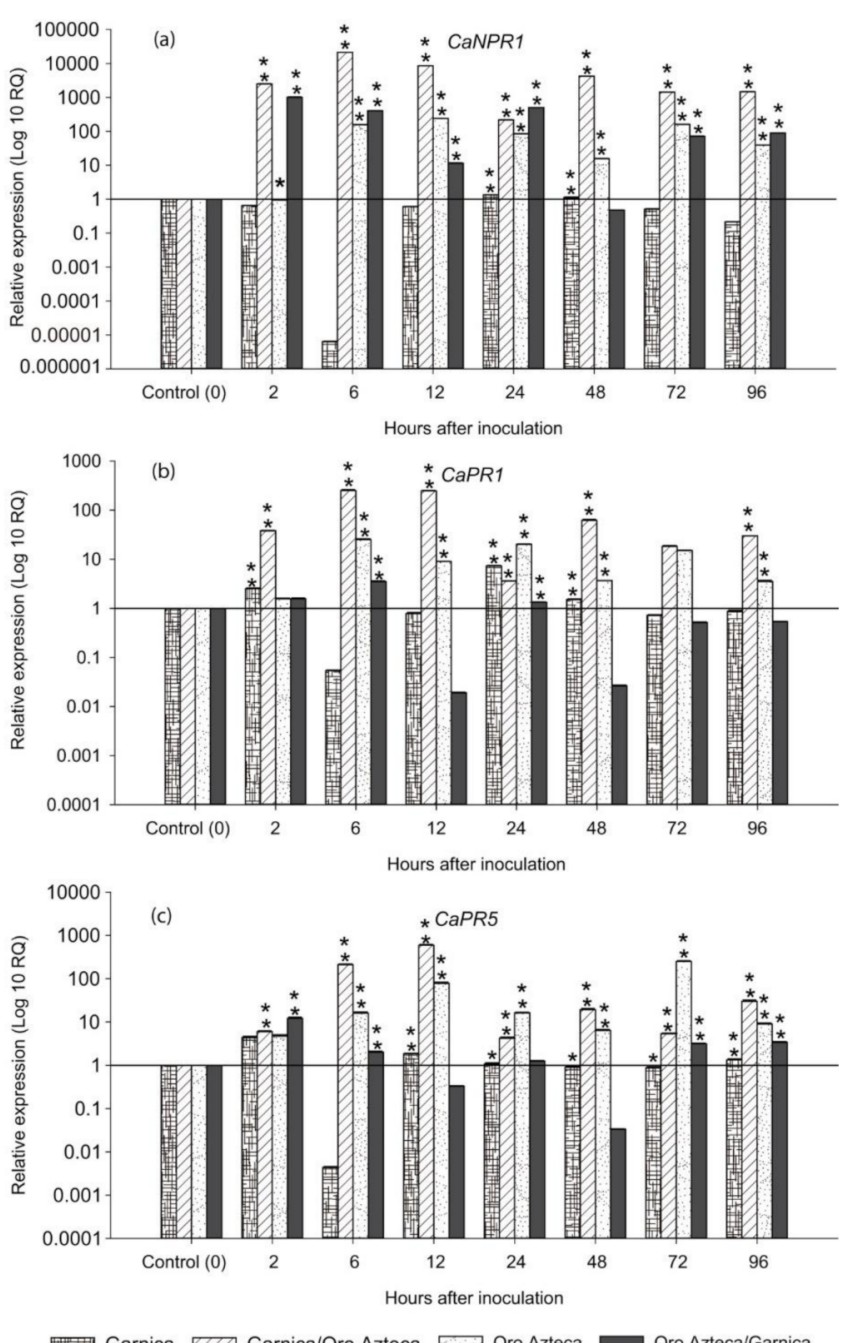

**Figure 4.** Relative expression of genes associated with the salicylic acid pathway, *CaNPR1* (**a**), *CaPR1* (**b**) and *CaPR5* (**c**), in ungrafted and grafted plants during 96 hai and using as control ungrafted plants. * There were no significant differences between the control and the treatment. ** There were significant differences between the treatments.

## 4. Discussion

### 4.1. Proliferation of Fungal Structures in Ungrafted and Grafted Plants

Comparing the four conditions of grafting studied, we found different patterns of fungal growth. In ungrafted Garnica plants, *H. vastatrix* growth was characterized by a higher percentage of fungal structures (anchors, HMC and haustoria) per infection point. On the pther hand, the ungrafted Oro Azteca. as a CLR-resistant cultivar, presented the lowest percentage of fungal structures [68]. Grafts had a different response than ungrafted plants, and Garnica grafted onto Oro Azteca showed less proliferation of the structures

compared to the ungrafted Garnica. The comparative condition of Garnica grafted onto Oro Azteca stands out, given that the percentage of anchors and HMC was similar to the ungrafted Oro Azteca and the graft of Oro Azteca onto Garnica (Figure 2). The same finding was observed when the incidence index of CLR was recorded at 236 dai (Table S4). Our results suggest that the response of the CLR-susceptible plants grafted on the CLR-resistant rootstock could decrease susceptibility to the pathogen. A similar behavior has been reported on the development of fungal structures among incompatible and compatible interactions of *H. vastatrix* with different varieties of *C. arabica* [55,58,59,68–70].

Reduced susceptibility to several pathogens by grafting has been previously described in other crops. In apples (*Malus pumila*), the use of rootstocks resistant to fire blight (*Erwinia amylovora*) in grafted plants in the Gala variety (susceptible to disease) showed a decrease in proliferation of the disease in the plant tissues [29,30]. Similar results have been obtained by evaluating the effect of the rootstock in the resistance to downy mildew (*Pseudoperonospora cubensis*) in cucumbers (*Cucumis sativus* L.), when grafted onto black seed squash (*Cucurbita ficifolia*) [33]. Another example can be observed in chili cv. Total (*Capsicum annuum*) susceptible to powdery mildew (*Leveillula taurica*), which showed a decreased presence of the pathogen on the leaves compared to the ungrafted susceptible plants when grafted on a cherry pepper rootstock (*Capsicum annuum* var. *cerasiforme*) resistant to disease [32]. The inter-specific grafting between watermelon (*Citrullus lanatus* L.) cv. Mickey Lee, susceptible to powdery mildew (*Podosperma xanthii*), and the bottle gourd (*Lagenaria sceraria* L.) rootstock significantly reduced the presence of the pathogen in the graft leaves [71]. Similarly, our results showed that for coffee, the use of rootstocks resistant to CLR could be a viable complementary tool to manage and control the disease.

*4.2. Changes in Plant Defense Responses Induced by the Rootstock*

In most interactions between plants and pathogens, the defense mechanisms start with the recognition of the pathogen, resulting in the start of the response followed by the activation of the defense genes [72,73]. In coffee, the *CaNDR1b* and *CaNBS-LRR* genes have been identified among the main recognition genes that participate in the activation of the pathogen-associated molecular pattern (PAMP)-triggered immunity (PTI) and effector-triggered immunity (ETI) defensive mechanisms [60,68,72,74].

Our results showed that grafted plants expressed the *CaNDR1b* gene significantly more than the susceptible cv. Garnica ungrafted plants, at all sampled times. The grafts Garnica/Oro Azteca showed outstanding expression levels of *CaNDR1b* at 6 and 12 hai. This could be an indicator that the rootstock is inducing changes in the expression of the *CaNDR1b* gene in the cv. Garnica scion, prompting a response in the levels even higher than those of the ungrafted CLR-resistant Oro Azteca. Thus, the recognition level of *H. vastatrix* in susceptible plants grafted onto resistant varieties would be similar to that which would be expected in resistant cultivars [68,75]. This leads us to consider that the rootstock influences some of the plant defense mechanisms. The change in gene expression of the *NDR1* family, by means of the rootstock, has also been described for *Vitis* sp. It was shown that gene expression levels, especially of genes involved in defense responses, in leaves of grafts of the Gaglioppo cultivar was dependent on the rootstock used [43].

In regards to the expression analysis of the *CaNBS-LRR* gene, we observed that compared to *CaNDR1b*, the grafted plants Garnica/Oro Azteca showed higher expression levels than ungrafted Garnica, only at 6, 12 and 48 hai. In apples, it was found that decreasing rates of necrosis caused by *E. amylovora* in grafted plants of the susceptible cv. Gala onto resistant G30 and MM.111 rootstock involved the expression of LRR pathogenesis recognition genes [29]. This could explain why the *CaNBS-LRR* expression was lower in leaves of the ungrafted Garnica plants during the proliferation of fungal structures in our study. In contrast, in grafted Oro Azteca/Garnica plants, the expression levels were lower than the control at 12, 24 and 48 hai, and the presence of fungal structures was slightly higher than in ungrafted CLR-resistant Oro Azteca plants. This shows that the susceptibility of the plant to pathogens can vary, depending on the characteristics of the

graft and rootstock used. For example, it was observed that a susceptible graft could affect the response to the soil borne pathogen *Ralstonia solanacearum* in an eggplant rootstock [76]. This would indicate that the expression of the *NBS-LRR* genes associated with the gene-for-gene resistance [77–80] in grafted plants could be mediated by the rootstock. This behavior was also observed in *Vitis sp.* [43,44,81]. Therefore, the increase in expression of the genes of the *LRR* family could be an indicator that CLR-resistant rootstock induce CLR-susceptible grafts, a similar defense response to that of an incompatible interaction.

For genes associated with the SA-response and -signaling, which is key in the defense mechanisms of *H. vastatrix* [68,82] and which is related to the triggering of the systemic acquired resistance (SAR) [68,83,84], we observed that the expression of gene *CaNPR1* in Garnica/Oro Azteca was higher than in ungrafted Oro Azteca in all the sampled times. Thus, the response to the rust attack would be expected to be similar, matching our observations on the proliferation of fungal structures (Figure 1). *NPR1* triggers the initial immune response through the expression of genes encoding the pathogenesis-related (PR) proteins *PR1* and *PR5* [85].

The expression level of *CaPR1* gene in the grafts Garnica/Oro Azteca was similar to that of gene *CaNPR1*, being higher than in the ungrafted plants in both cultivars in almost all sampled times. However, the *CaPR1* expression in grafted plants Oro Azteca/Garnica was lower than in all other conditions in almost all of the sampled times. This could indicate that Garnica cultivar as rootstock could affect the resistance ability of Oro Azteca when grafted onto it, as it was observed in scion of *Cedrela* species grafted onto *Toona ciliate*, that are attacked by *Hypsipyla robusta* [34]. However, the observations on the proliferation of the fungal structures in foliar tissue showed differences between both grafted conditions (Figure 1). The expression level of this gene in leaves of the cv. Gaglioppo grape was also dependent on the rootstock used [43].

The expression of *CaPR5*, the other SA-responsive gene that we analyzed, showed changes as an effect of the rootstock, since, in both cases, these plants behaved differently to what was expected. The grafted plants Oro Azteca/Garnica had the lowest expression of all treatments at 12, 24 and 48 hai. Thus, it is possible to observe an effect at the level of the gene defense regulation and expression processes, induced by the rootstock. Albert et al. [32] evaluated the use of *PR1* and *PR5* genes as markers of the transmission of resistance facilitated by grafting. Their results showed that after 45 days of inoculation, the expression levels were similar between the grafted plants of the chili cv. Total, susceptible to *L. taurica* onto the "Cereza chili" resistant cv. Szentesi, and those of the ungrafted Szentesi cultivar. Therefore, the expression levels in *CaPR1* and *CaPR5* in both grafting conditions could indicate that the rootstock modifies the response to rust in the grafted plants.

## 5. Conclusions

Our results show that the rootstock influences the defense response to infection by *H. vastatrix*, responsible for the CLR, thus, being able to confer resistance to susceptible cultivars when they are CLR-resistant rootstocks. It is important to highlight that the main regulator of the SA-dependent pathway, *CaNPR1* (which is linked to SAR), was highly expressed in the evaluated grafts and in the resistant Oro Azteca. However, more in-depth research needs to be conducted on the interaction between coffee rootstock and scion, especially regarding systemic long-distance signaling and communication between them. Our findings broaden the spectrum of options to manage CLR in an integrated way, thus contributing to the reduction of chemical fungicide use.

**Supplementary Materials:** The following are available online at https://www.mdpi.com/article/10.3390/agronomy11081621/s1, Figure S1: Schematic design for obtaining reciprocal grafts of cv. Garnica and cv. Oro Azteca, susceptible (S) and resistant (R) to CLR respectively, Figure S2: Symptoms caused by *H. vastatrix* infection in ungrafted and grafted cv. Garnica (G) and cv. Oro Azteca (OA) plants. Table S1: Significance of the effect of time since infection on the number of fungal structures in each experimental condition (Kruskal-Wallis test), Table S2: Comparison of average ranges obtained by the Kruskal-Wallis variance test of the fungal structure counts at each time for

the four experimental conditions [86], Table S3: Frequency of CLR symptomatic plants at 30, 60, 100, 130, 165, 208 and 236 days after inoculation, Table S4: Analysis of the incidence and severity (percentage of damaged leaf area and number of sporulated pustules) of rust infection at 236 days after inoculation [87].

**Author Contributions:** Conceptualization, E.C.-B., M.M.-R. and A.M.-B.; methodology, E.C.-B., G.C., L.V., F.O.-E., M.M.-R. and A.M.-B.; formal analysis, E.C.-B., G.C., L.V., M.M.-R. and A.M.-B.; investigation, E.C.-B., F.O.-E. and A.M.-B.; resources, M.M.-R. and A.M.-B.; data curation, E.C.-B.; writing—original draft preparation, E.C.-B. and A.M.-B.; writing—review and editing, E.C.-B., G.C., L.V., M.M.-R. and A.M.-B.; supervision, M.M.-R. and A.M.-B.; project administration, M.M.-R. and A.M.-B.; funding acquisition, M.M.-R. and A.M.-B. All authors have read and agreed to the published version of the manuscript.

**Funding:** This research was funded by Consejo Nacional de Ciencia y Tecnología: CB-2014, 242999 and LN-2017, 280505, by INECOL: Institutional funding 2003010806. The APC was funded by ENES Morelia, UNAM.

**Institutional Review Board Statement:** Not applicable.

**Informed Consent Statement:** Not applicable.

**Conflicts of Interest:** The authors declare no conflict of interest.

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
