# Peer review of "Defense Response to Hemileia vastatrix in Susceptible Grafts onto Resistant Rootstock of Coffea arabica L."

_agronomy, doi:10.3390/agronomy11081621_

Round 1

Reviewer 1 Report

In this study, conducted coffee grafting with a susceptible and a resistant cultivar and examined infection level of a coffee pathogen, coffee leaf rust (CLR), as well as expression of genes associated with pathogenesis and salicylic acid pathway. As results, higher gene expression was observed in the resistant grafts. I think that the data presented in this study is reasonable and provides meaningful information in the corresponding field. I only provide minor comments.

  1. Showing photographs of haustoria, anchor and HMC is helpful to catch information practically.

  1. Line 328-330, the authors describe “we observed that the graft Oro Azteca onto Garnica showed a highest presence of fungal structure that than the ungrafted Oro Azteca”. However, both ungrafted and grafted Oro Azteca seems to have a similar level of infection in Figures 1 and 2. Are there any significant difference? Please clarify.

  1. Line 428, “rootstock and Scion” should be “rootstock and scion”.

Author Response

Response to Reviewer 1 Comments and Suggestions

- In this study, conducted coffee grafting with a susceptible and a resistant cultivar and examined infection level of a coffee pathogen, coffee leaf rust (CLR), as well as expression of genes associated with pathogenesis and salicylic acid pathway. As results, higher gene expression was observed in the resistant grafts. I think that the data presented in this study is reasonable and provides meaningful information in the corresponding field. I only provide minor comments. 

Thanks a lot for the constructive comments, we are convinced that an integrated management of CLR that includes the use of resistant rootstocks will contribute to reduce the damage caused by the disease at a global level.

1. Showing photographs of haustoria, anchor and HMC is helpful to catch information practically.

We have included a new figure (Fig. S2) which shows representative images of cytological analysis of inoculated leaves.

2. Line 328-330, the authors describe “we observed that the graft Oro Azteca onto Garnica showed a highest presence of fungal structure that than the ungrafted Oro Azteca”. However, both ungrafted and grafted Oro Azteca seems to have a similar level of infection in Figures 1 and 2. Are there any significant difference? Please clarify.

We agree with the reviewer, the sentence and information was confuse. The sentence has been modified (please refer to the manuscript L336) “Garnica grafted onto Oro Azteca showed less proliferation of the structures compared to the ungrafted Garnica”.

3. Line 428, “rootstock and Scion” should be “rootstock and scion”.

We have attended these and some other typing mistakes, please refer to the manuscript L433.

Reviewer 2 Report

1. The evidence and arguments are presented well and addresses the main questions. Researchers confirmed that the grafted plants are more resistant to the fungal infection than the ungrafted plants. It has been confirmed by gene expression profiles for pathogen-recognition genes and salicylic acid pathways genes. Despite that, authors should include images of microscopical samples. This will help strengthen the findings.

2. With regards to novelty, this does not constitute a novel experiment, as the same authors have already published the work with coffee leaf rust in the year 2020, but in the current experiment, they have been able to extend their findings to plants that were produced by grafting resistant with sensitive varieties. The authors should include more details about the procedure used in making grafts. An image or schematic diagram can be included. Authors are also encouraged to include images of plants. Did the offspring of the grafted plants show the same level of resistance to the fungal infection?

3. The manuscript is written well, but minor spelling corrections are needed. In contrast to previous published studies, there were differentially expressed salicylic pathway genes in grafted plants and the ungrafted plants. This may be a positive step towards grafting as a method of crop protection.

Line 31.. Cite references that indicate pests are a key factor in spreading CLR

Line 36 to 38 ,, sentences can be rewritten

Please include microscopical images of disease progression in the infected leaves

Schematic diagram or figures representing, how does the grafting was made between the varieties

Author Response

Response to Reviewer 2 Comments and Suggestions

1. The evidence and arguments are presented well and addresses the main questions. Researchers confirmed that the grafted plants are more resistant to the fungal infection than the ungrafted plants. It has been confirmed by gene expression profiles for pathogen-recognition genes and salicylic acid pathways genes. Despite that, authors should include images of microscopical samples. This will help strengthen the findings.

We appreciate the comment. We have included a new figure (Fig. S2) which shows representative images of cytological analysis of inoculated leaves.

2. With regards to novelty, this does not constitute a novel experiment, as the same authors have already published the work with coffee leaf rust in the year 2020, but in the current experiment, they have been able to extend their findings to plants that were produced by grafting resistant with sensitive varieties. The authors should include more details about the procedure used in making grafts. An image or schematic diagram can be included. Authors are also encouraged to include images of plants. Did the offspring of the grafted plants show the same level of resistance to the fungal infection?

Our previous published work was performed before to the present study, in order to corroborate the supposed resistance of cv. Oro Azteca. The infection assays presented by Couttolenc-Brenis et al, 2020, were performed locally in detached leaf disks, inside of humid boxes. In our present study, infection was done with ungrafted or grafted whole-plants. Regarding the grafting procedure, we have added some additional details in the Materials and Methods section, please refer to manuscript L106-109. Additionally, we have generated an illustrating scheme of the grafting design, as suggested (Fig. S1). A representative image of the infected plants at 120 days after infection is showed in the new Figure S3. Although we monitored the effect of CLR up to 240 days after infection, we did not evaluated the possible resistance conferred on the offspring; We appreciate the understanding of the reviewer.

3. The manuscript is written well, but minor spelling corrections are needed. In contrast to previous published studies, there were differentially expressed salicylic pathway genes in grafted plants and the ungrafted plants. This may be a positive step towards grafting as a method of crop protection.

We have corrected several grammatical and typing mistakes, we hope we have attended all.

- Line 31.. Cite references that indicate pests are a key factor in spreading CLR.

The meaning of our phrase was misunderstood because our incomplete redaction. The sentence was reformulated into: “…controlling and managing diseases and pests is key to preserving the income of producers that depend on this crop. Coffee leaf rust (CLR) is one of the diseases that has most impacted global coffee production in recent years…”. Please refer to manuscript L31.

- Line 36 to 38 ,, sentences can be rewritten.

The sentence was reformulated, please refer to manuscript L36-38.

- Please include microscopical images of disease progression in the infected leaves.

We have added a new figure (Fig. S2) which shows representative images of cytological analysis of inoculated leaves.

- Schematic diagram or figures representing, how does the grafting was made between the varieties.

We have added some methodological details, please refer to the manuscript “Materials and Methods” section, L106-109. We have also generated an illustrating scheme of the grafting design, as suggested (Fig. S1).